# STREAMLINING GENERATIVE MODELS FOR STRUCTURE-BASED DRUG DESIGN

## ABSTRACT

Generative models for structure-based drug design (SBDD) aim to generate novel 3D molecules for specified protein targets *in silico*. The prevailing paradigm focuses on model expressivity - typically with powerful Graph Neural Network (GNN) models - but is agnostic to binding affinity during training, potentially overlooking better molecules. We address this issue with a two-pronged approach: learn an economical surrogate for affinity to infer an unlabeled molecular graph, and optimize for labels conditioned on this graph and desired molecular properties (e.g., QED, SA). The resulting model FastSBDD achieves state-of-the-art results as well as streamlined computation and model size (up to 1000x faster and with 100x fewer trainable parameters compared to existing methods), paving way for improved docking software. We also establish rigorous theoretical results to expose the representation limits of GNNs in SBDD contexts and the generalizability of our affinity scoring model, advocating more emphasis on generalization going forward.

## 1 INTRODUCTION

Identifying *ligands* that bind well to a protein target and have desired properties is a key challenge of Structure-based drug design (SBDD), while experimental determination of binding is costly. Recently deep generative models have proposed to accelerate SBDD by suggesting good candidate molecules from learned conditional ligand distribution given a target protein from experimental or *in silico* binding observations (Vamathevan et al., 2019; Bilodeau et al., 2022). These approaches include autoregressive models (Peng et al., 2022; Luo et al., 2021a), variational autoencoders (Ragoza et al., 2022), reinforcement learning (Li et al., 2021), and diffusion models (Schneuing et al., 2022).

However, existing methods are oblivious to binding affinity as they seek to match the observed ligand distribution, which often includes molecules with varying affinities. Furthermore, they optimize jointly atom types and 3D coordinates with powerful GNN encoders with typically expensive training and sampling procedures. We address these issues with a novel two-phase generative method *FastSBDD* that draws inspiration from the success of performance-aware methods in other domains (Joachims, 2005), and directly optimizes for the metrics of interest, i.e. properties and binding affinity.

Specifically, we first learn a cheap affinity surrogate to infer an unlabeled molecular graph, and then optimize its atom labels and coordinates. We demonstrate the versatility of our approach with three versions of FastSBDD tailored to the problems of (i) drug repurposing, which generates samples from a known ligand set; (ii) drug generation, which generates novel unseen ligands; and (iii) property optimisation, which seeks to control both molecular properties and binding. FastSBDD achieves SBDD state of the art with up to 1000x speed up and 100x fewer parameters.

A broader objective, and contribution, of this work is to foster theoretical underpinnings for SBDD. We take two important steps in this pursuit. First, we expose the limits on the expressivity of GNNs to distinguish distinct molecules conditioned on protein targets. Next we prove generalization bounds for our binding surrogate to ground its strong empirical performance. While such results on representational limits and generalization ability are known for GNN models in unconditional settings (Garg et al., 2020; Joshi et al., 2023; Ju et al., 2023), to our knowledge, these are the first results in conditional drug discovery contexts.

## 1.1 OUR CONTRIBUTIONS

We summarize our main contributions below. Specifically, we

1. **(Conceptual)** introduce metric-aware optimization for generative SBDD settings;

2. **(Methodological)** propose FastSBDD as a streamlined generative framework that optimizes for docking scores and physicochemical properties;

3. **(Empirical)** show that FastSBDD significantly outperforms existing methods with orders of magnitude fewer parameters and faster runtime; and

4. **(Theoretical)** analyze representational challenges of GNNs in SBDD contexts, and establish that FastSBDD generalizes well with respect to predicting scores for new target-ligand pairs.

## 2 PRELIMINARIES AND RELATED WORK

**Deep (Geometric) Learning for Drug Design**. Deep learning has enabled progress on various facets of drug design across molecular generation (Jin et al., 2018; Zang & Wang, 2020; Verma et al., 2022; Shi et al., 2020; Luo et al., 2021b; Hoogeboom et al., 2022; Igashov et al., 2022), molecular optimization (Jin et al., 2019) and drug repurposing (Pushpakom et al., 2019). It has also accelerated advancements in protein folding and design (Jumper et al., 2021; Wu et al., 2022b; Ahdritz et al., 2022; Verkuil et al., 2022; Hie et al., 2022; Shi et al., 2023; Watson et al., 2022; Ingraham et al., 2022; Wu et al., 2022a; Verma et al., 2023) and docking (Stärk et al., 2022b; Ganea et al., 2022; Corso et al., 2023). We refer the reader to the surveys by Chen et al. (2018) and Pandey et al. (2022).

Graph Neural Networks (GNNs) (Kipf & Welling, 2017; Hamilton et al., 2017; Veličković et al., 2018) have recently emerged as a workhorse for modeling data, such as molecules, that can be represented as graphs. They can be extended to encode important symmetries occurring in geometric graphs (Satorras et al., 2021), so have found success across diverse tasks in the drug design pipeline including docking (Corso et al., 2023) and molecular property prediction (Stärk et al., 2022a).

**Molecular properties**   The goal of SBDD is to generate drug candidates with high binding affinity to a target protein, while controlling for other properties. We list below common molecular properties, calculated in practice using the RDKit software (Landrum et al., 2020), that are relevant to drug design (Bilodeau et al., 2022; Peng et al., 2022):

- Binding affinity – the strength of the natural connection between the ligand and the protein. A common surrogate for affinity is the Vina score (Alhossary et al., 2015).

- LogP – logarithm of the partition coefficient, which is a solubility indicator.

- QED – quantitative estimate of drug-likeness, a mixture of properties correlated with drugs.

- SA – Synthetic accessibility, an estimate of how easy it would be to synthesize the molecule.

- Lipinski's Rule of Five – A heuristic about whether a molecule would be an active oral drug.

**Setting**. We view proteins $P$ as labelled graphs: each pocket $G = (V, E)$ consists of nodes $V = \{v_1, \ldots, v_N\}$ with $v_i$ representing atoms, and edges $E \subseteq V \times V$. Each node $v = (a, \mathbf{s})$ is associated with an atom type $a \in \mathcal{A} = \{\mathtt{C}, \mathtt{N}, \mathtt{O}, \ldots\}$ and 3D coordinates $\mathbf{s} \in \mathbb{R}^3$, and each edge $e \in E$ with a bond type from $\{1, 2, 3\}$. We represent the molecule $M$ analogously as another graph. We often include superscripts $G^P$ and $G^M$ to distinguish between protein and molecule graphs. We also distinguish between a labelled molecular graph $G^M$ and unlabelled one $U^M$. The unlabelled graph contains information about the number of nodes and connectivity between them, but no information about atom types or 3D coordinates.

The problem of SBDD can be posed as modeling the conditional distribution $p(G^M | G^P)$. The main challenge of SBDD is to generate candidate ligands with high binding affinity to a target protein.

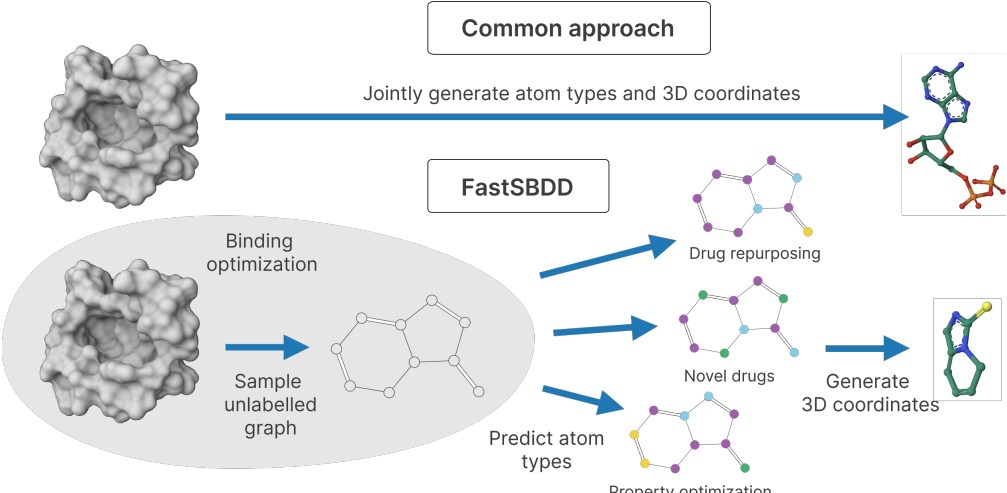

Figure 1: **Comparison of FastSBDD to common approaches**. Top: SBDD approaches commonly learn to approximate the data distribution of atom types and 3D coordinates conditioned on the protein pocket. Bottom: FastSBDD first generates the unlabelled graph explicitly optimized for binding affinity. Then it predicts atom types using different strategies designed for solving different tasks independently of the protein pocket. Finally, it generates a 3D configuration.

## 3 FAST STRUCTURE-BASED DRUG DESIGN

### 3.1 DECOUPLING THE UNLABELLED MOLECULAR GRAPH FROM ATOM TYPES

One of the main innovations underlying our framework is the separation of molecular representation into the unlabeled molecular graph and atom types. As a motivation, we first investigate how Vina binding scores behave when we modify the atom types and coordinates, while keeping the underlying graph intact. After making these adjustments, we measured the Vina scores of the modified molecules and compared them to the scores of their unaltered counterparts.

Remarkably, our analysis showed a strong coefficient of determination, with an $R^2$ value of 0.76, between the scores of the original molecules and their modified counterparts. This significant result highlights that a considerable portion of the variability in binding affinity, as quantified by the Vina score, can be attributed to the information contained within the unlabeled molecular graph itself, even before incorporating atom-specific details. Please see Appendix A for details.

It is important to recognize that these findings pertain to the Vina software and not the intrinsic biological process of binding. While Vina is the most prevalent approximation, it remains just that – an approximation. Nonetheless, upon conducting similar experiments with Gnina (McNutt et al., 2021), a reportedly significantly more accurate binding approximation than Vina, we found analogous results. Please see Appendix A.4 for details. This suggests that the observed behavior is not solely an artifact of Vina but may be more widespread.

Inspired by these findings, we propose a new disentangled model FastSBDD (see Figure 1). We first generate the high-binding unlabelled graph structures $U^M$, and generate atom types $\mathbf{a}^M$ independently of the protein. The model is defined by a disentangled conditional distribution

$$p(G^M|G^P) \stackrel{\text{def}}{=} p(\mathbf{a}^M, \mathbf{s}^M|U^M, \cancel{G^P})p(U^M|G^P), \tag{1}$$

where we assume atoms $\mathbf{a}^M$ are conditionally independent of the protein pocket $G^P$ given the unlabelled graph structure $U^M$. The generative model thus reduces to two independent components: unlabelled graph sampler and atom sampler. We discuss them below.

### 3.2 UNLABELLED GRAPH SAMPLER

Typically, one would train the unlabelled graph model $p(U^M|G^P)$ to match the empirical distribution. However, we hypothesize that observed ligand-protein complexes contain examples with suboptimal

binding. We first learn a binding surrogate $g_\theta$ to predict binding affinity from unlabelled graph alone,

$$g_\theta(U^M, G^P) \approx \text{Vina}(G^M, G^P). \tag{2}$$

Next, we use $g_\theta$ to score unlabelled graphs from a repository of molecules, and choose those with the best predicted affinities. We define a generative distribution

$$p_\theta(U^M|G^P) = \text{UNIFORM}\Big(\big\{U^M \in \mathcal{U}^M \mid v_{\min} \leq g_\theta(G^P, U^M) \leq v_{\max}\big\}\Big), \tag{3}$$

where $\mathcal{U}^M$ is a database of unlabelled graph structures, and $v_{\min}, v_{\max}$ are threshold hyperparameters. We set $v_{\min}$ and $v_{\max}$ to be the 5th and 10th percentiles of all predictions for $\mathcal{U}^M$, and filter out the best 5% of the scores to avoid outliers. As $\mathcal{U}^M$, we used the ZINC250k dataset (Irwin et al., 2012) instead of the CrossDocked2020 (Francoeur et al., 2020), which only contains 8,000 unique ligands.

We note that instead of sampling from a database, one can define a deep generative model over unlabelled graphs. However, given that there is substantial variability in the molecular space even with predefined unlabelled graphs, we have not investigated this direction.

**Architecture of the scoring model $g_\theta$**   To parametrize the binding surrogate $g_\theta$, we chose the E(3)-equivariant graph neural network (EGNN) (Satorras et al., 2021), which computes a learnable representation of the protein pocket. This protein pocket representation is concatenated with features describing the ligand's unlabelled graph. We chose as features (i) number of nodes, (ii) number of rings, (iii) number of rotatable bonds, and (iv) graph diameter. The features are passed through an MLP with ReLU activations. Remarkably, these four simple features already perform well, and more sophisticated methods like GNNs did not bring significant improvements. See Appendix B for details.

**Training data for $g_\theta$**   To train the binding affinity predictor $g_\theta$, we need pairs of protein-ligand pairs with their affinity estimates. We chose not to use the CrossDocked2020 dataset, the gold standard for SBDD, for the following reasons: 1) even though it contains 100,000 protein-ligand pairs, there are only around 8,000 unique ligands, which can be insufficient for generalisation, 2) for a given protein pocket there may exist ligands with better affinity than what is present in the data and 3) for the scoring model to assign good values only to good structures, it needs to be trained also on examples with poor binding affinity that are not available.

To address the above, we construct a dataset ourselves. We sample 1000 diverse ligand-protein complexes from CrossDocked2020, and couple 50 additional random molecules from the ZINC250k. This results in 51,000 pairs, for which we compute the Vina ground-truth binding affinity. The scoring model $g_\theta$ is trained with stochastic gradient descent to minimize the mean squared error between prediction $g$ and Vina scores. See Appendix B.1 for details.

**Scoring model detects optimal molecule size**   To understand what the scoring learned, we visualized its predictions as a function of the size of the molecule. We notice that protein pockets fall under two cases: either there is an optimal size of the ligand (Figure 2a) or there is a monotonic relationship favouring larger ligands (Figure 2b). We hypothesize the model learnt the optimal ligand size per pocket, but for some pockets this size is larger than the largest available molecule (38 atoms) in the ZINC250k dataset.

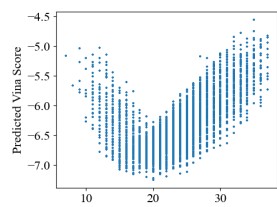
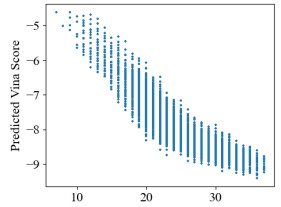

(a) Protein pocket with optimal ligand size $\approx 20$

(b) Protein pocket with optimal ligand size $\geq 38$

Figure 2: **Optimal ligand size has two modes**. Representative examples showing that the scoring model learns the optimal ligand size which is different for different proteins.

### 3.3   ATOM SAMPLER

The atom sampler $p(\mathbf{a}^M, \mathbf{s}^M|U^M)$ generates atom types and coordinates of each atom. As mentioned in Section 3.1, the precise 3D conformation has marginal effect on the Vina score, and therefore we simply generate any valid 3D configuration with RDKit. We choose a factorisation

$$p(\mathbf{s}^M, \mathbf{a}^M|U^M) = p(\mathbf{s}^M|\mathbf{a}^M, U^M)p(\mathbf{a}^M|U^M), \tag{4}$$

where $p(\mathbf{s}^M|\mathbf{a}^M, U^M)$ is an off-the-shelf conformation generation model available in RDKit. In practice, the 3D configuration of the ligand must be in the vicinity of the protein pocket in order to correctly compute its binding affinity using Vina. We therefore train a separate model which predicts the center of mass of the ligand based on the protein pocket, and use it to center the conformers. See Appendix C for more details.

We propose three different approaches to modeling the atom type sampler $p(\mathbf{a}^M|U^M)$ that are tailored for solving different problems.

### 3.3.1 DRUG REPURPOSING

We begin with a model FastSBDD-$\mathcal{DR}$ for drug repurposing, i.e. sampling from known molecules that are synthesizable. We define the atom sampler to follow the empirical distribution of known ligands with a given unlabelled graph $U^M$:

$$p_{\text{DR}}(\mathbf{a}^M|U^M) = \text{UNIFORM}\left(\{\mathbf{a}^M|(\mathbf{a}^M, U^M) \in \mathcal{D}\}\right), \tag{5}$$

where $(\mathbf{a}^M, U^M)$ denotes a molecule with unlabelled graph $U^M$ and atoms $\mathbf{a}^M$, and $\mathcal{D}$ is a molecular database. In our experiments, we used ZINC250 dataset (Irwin et al., 2012).

### 3.3.2 NOVEL DRUG GENERATION

FastSBDD can also be applied for generation of novel drugs. We adapt an existing deep hierarchical generative model MoFlow (Zang & Wang, 2020), consisting of an unlabelled graph model $f_U$ and a conditional atom model $f_{\mathbf{a}|U}$. The atom model induces a conditional distribution $p_{f_{\mathbf{a}|U}}(\mathbf{a}^M|U^M)$, which we use as our atom sampling distribution for novel drug generation:

$$p_{\mathcal{ND}}(\mathbf{a}^M|U^M) = p_{f_{\mathbf{a}|U}}(\mathbf{a}^M|U^M). \tag{6}$$

We use a pre-trained MoFlow with no further training or fine-tuning, so omit them from the trainable parameter count of the result (Table 1). We denote this model variant as FastSBDD-$\mathcal{ND}$.

### 3.3.3 PROPERTY OPTIMIZATION

Finally, we show that our approach allows for joint optimization of molecular properties and binding affinity. As an illustrative example, we follow Gómez-Bombarelli et al. (2018) and choose $5 \cdot \text{QED} + \text{SA}$ as the property mixture to optimize. We first sample 50 candidates for atom assignments using $\mathbf{a}^M \sim p_{\mathcal{ND}}(\mathbf{a}^M|U^M)$, and choose the one with the best value of the desired property. We emphasize that this optimization is performed with respect to atom types, while keeping the unlabelled graph structure $U^M$ fixed. We denote this model FastSBDD-$\mathcal{PO}$.

We also consider optimization in the latent space of the atom type generative model. Since the atom type generative model of Section 3.3.2 is a normalizing flow, we can compute a latent representation of each atom type assignment. We use the pre-trained model for predicting molecular properties based on the latent representation of a molecule, and used its gradients to perform optimization in the latent space. This approach has the benefit of being faster as it does not require passing through the flow model at each iteration. In our experiments however this yielded worse results than the random search, suggesting suboptimal performance of the property predictor model.

## 4 RESULTS

**Data** We follow related work of Peng et al. (2022); Schneuing et al. (2022) and use the CrossDocked2020 dataset (Francoeur et al., 2020) for evaluating the models in our experiments. We use the same train-test split, which seperates protein pockets based on their sequence similarity computed with MMseqs2 (Steinegger & Söding, 2017) and contains 100,000 train and 100 test protein-ligand pairs.

We follow the evaluation procedure described in Peng et al. (2022). For each of 100 test protein pockets, we

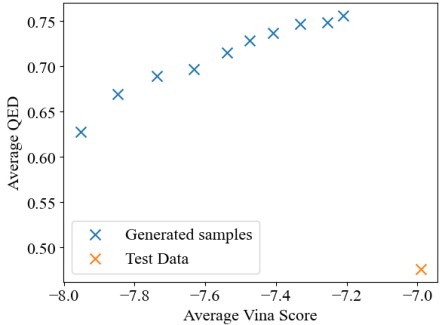

Figure 3: **Binding Affinity vs QED tradeoff**. Our models allow for controlling the binding affinity of predictions.

Table 1: **FastSBDD finds high-quality drug candidates up to 1000x faster than competing methods**. Comparison of multiple properties of the CrossDocked2020 test data and other generation methods. Results for DiffSBDD were taken from Schneuing et al. (2022), where "High Affinity" was not reported. To report results for TargetDiff, we used samples provided by the authors. Reported runtime was taken from their paper, where standard deviation was not reported. † denotes method run exclusively on CPU. The #Params column refers to the number of trainable parameters.

| | Vina Score (kcal/mol, ↓) | High Affinity (↑) | QED (↑) | SA (↑) | Diversity (↑) | #Params (↓) | Time (s, ↓) |
|---|---|---|---|---|---|---|---|
| Test set | -6.99 ± 2.16 | - | 0.48 ± 0.21 | 0.73 ± 0.14 | - | - | - |
| DiffSBDD | -6.58 ± 2.06 | - | 0.50 ± 0.15 | 0.34 ± 0.09 | **0.73** ± 0.11 | 3.5M | 1634 ± 769 |
| Pocket2Mol | -7.10 ± 2.56 | 0.55 ± 0.31 | 0.57 ± 0.16 | 0.74 ± 0.13 | **0.72** ± 0.16 | 3.7M | 2504 ± 220 |
| TargetDiff | -6.91 ± 2.25 | 0.52 ± 0.32 | 0.48 ± 0.20 | 0.58 ± 0.13 | **0.72** ± 0.09 | 2.5M | 3428 ± NA |
| FastSBDD–$\mathcal{DR}$ | **-7.82** ± 1.47 | **0.72** ± 0.36 | 0.66 ± 0.15 | **0.78** ± 0.08 | 0.68 ± 0.05 | **23K** | **1.8**$^\dagger$ ± 0.4 |
| FastSBDD–$\mathcal{ND}$ | **-7.78** ± 1.47 | **0.71** ± 0.34 | 0.61 ± 0.18 | 0.69 ± 0.09 | 0.68 ± 0.06 | **23K** | 3.9$^\dagger$ ± 0.9 |
| FastSBDD–$\mathcal{PO}$ | **-7.98** ± 1.46 | **0.75** ± 0.35 | **0.80** ± 0.10 | 0.73 ± 0.08 | 0.66 ± 0.06 | **23K** | 115$^\dagger$ ± 11 |

sample 100 molecules using each of our sampling strategies described in this section. As evaluation metrics we report 1) **Vina Score** estimating binding affinity (Alhossary et al., 2015), 2) **High Affinity**, which is the percentage of generated molecules with binding affinity at least as good as ground truth, 3) **QED**, a quantitative estimate of druglikeness, 4) **SA**, synthetic accesibility, 5) **Diversity** defined by average Tanimoto dissimilarity for the generated molecules in the pocket, 6) **#Params**, the number of trainable parameters and 7) **Time** to generate 100 molecules in seconds. We include more molecular metrics in Appendix D. As baselines, we use three recent methods for SBDD: Pocket2Mol (Peng et al., 2022), DiffSBDD (Schneuing et al., 2022) and TargetDiff (Guan et al., 2023).

We report the results in Table 1. Clearly, all our approaches significantly outperform the previous state of the art, Pocket2Mol while being 20-1000x faster despite being run solely on CPU (and not on GPU unlike the baselines). Additionally, our method has only 23k trainable parameters, which is two orders of magnitude less than the baselines. We also note a decrease in diversity compared to these baselines: this is expected given that our model uses known unlabelled graphs during sampling.

**Trade-off between QED and Binding Affinity prediction**    As seen in Equation (3), our method allows for controlling the binding affinity of the generated ligands. We therefore investigated the performance of the model when we vary the $v_{\min}, v_{\max}$ thresholds. Specifically, we evaluated 10 different versions of the unlabelled graph sampler with $(v_{\min}, v_{\max})$ set to $(q_0, q_5), (q_5, q_{10}), \ldots, (q_{45}, q_{50})$, where $q_k$ is the $k^{th}$ percentile of the predictions for $\mathcal{U}^M$ (note that these percentiles depend on the protein pocket). We generated atom types using the repurposing model described in section 3.3.1. We note a trade-off between QED and Binding Affinity of these models, but all of them are still better on both of these criteria than the samples from the CrossDocked2020 dataset. We show this in Figure 3.

**The model generalizes across molecular datasets**    In our experiments we have used the ZINC250k dataset (Irwin et al., 2012) to train the scoring model (2), to sample unlabelled graphs (3) and to sample atom types in the drug repurposing model (5). We now check whether the model generalizes across datasets. To do that, we evaluate the FastSBDD–$\mathcal{DR}$ model with, but with the ChEMBL dataset (Mendez et al., 2018; Davies et al., 2015) without retraining the scoring model. We found that the model performs very similarly in terms of binding affinity. See Table 2 for details. We discuss model's generalization ability from a theoretical perspective in Section 5.2.

We now move on to studying the representational challenges and generalization of models for SBDD.

## 5 THEORETICAL ANALYSIS

We showed empirically that very simple representations of the unlabelled graph perform remarkably well in terms of predicting binding affinity, and allow for significantly streamlined computation compared to much larger and more sophisticated models. Here we bring attention to the representational issues concerning large models. We show that despite adding considerable complexity, such models might have weaknesses in terms of what they can represent. While these limitations can possibly be circumvented with even more complex models, the expense in terms of their weaker

Table 2: **The model generalizes to other molecular datasets.** Comparison of the drug repurposing model with different sets of molecular structures. Both use the scoring model trained on the ZINC250k dataset (Irwin et al., 2012), but FastSBDD–$\mathcal{DR}$ (ZINC) uses ZINC250k also during sampling, while FastSBDD–$\mathcal{DR}$ (ChEMBL) uses ChEMBL (Mendez et al., 2018; Davies et al., 2015).

| | Vina Score (kcal/mol, ↓) | High Affinity (↑) | QED (↑) | SA (↑) | Lipinski (↑) | LogP |
|---|---|---|---|---|---|---|
| Test set | $-6.99 \pm 2.16$ | - | $0.48 \pm 0.21$ | $0.73 \pm 0.14$ | $4.34 \pm 1.14$ | $0.89 \pm 2.73$ |
| FastSBDD–$\mathcal{DR}$ (ZINC) | $-7.82 \pm 1.47$ | $0.72 \pm 0.36$ | $0.66 \pm 0.15$ | $0.78 \pm 0.08$ | $4.99 \pm 0.05$ | $2.97 \pm 1.37$ |
| FastSBDD–$\mathcal{DR}$ (ChEMBL) | $-8.03 \pm 1.56$ | $0.76 \pm 0.33$ | $0.56 \pm 0.15$ | $0.79 \pm 0.08$ | $4.99 \pm 0.12$ | $3.48 \pm 0.95$ |

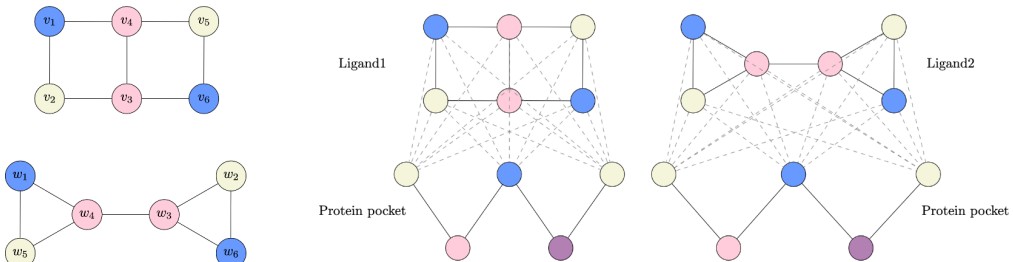

Figure 4: **Some ligand graphs cannot be distinguished by LU-GNNs even with additional protein context** Left: Construction for Lemma 1; Two non-isomorphic graphs differing in all properties stated in Lemma 1, but for which LU-GNNs produce identical embeddings. Right: Complex graphs constructed by joining the ligand graphs with the same protein graph remain identical whenever ligand graphs cannot be differentiated.

generalization ability (as known from learning theory results) could be excessive. Simpler, and likely less expressive, models like the ones considered in our work on the other hand might offer a better expressivity-generalization tradeoff and thus might be preferable from a theoretical perspective.

## 5.1 REPRESENTATION LIMITS OF GNNS IN SBDD

It is known that there exist graphs representing distinct molecular structures that cannot be distinguished using message passing graph neural networks (MPGNNs) (Sato, 2020; Garg et al., 2020). Following the notation in Garg et al. (2020), we consider *Locally Unordered* GNNs (LU-GNNs), which we summarize as follows. The updated embedding $h_v^{(l)}$ of node $v$ at layer $l$ is defined as

$$m_{u \to v}^{(l-1)} = \phi(h_u^{(l-1)}, e_{uv}); \tilde{h}_v^{(l-1)} = \text{AGG}\{m_{u \to v}^{(l-1)} \mid u \in N(v)\}; h_v^{(l)} = \text{COMBINE}\{h_v^{(l-1)}, \tilde{h}_v^{(l-1)}\},$$

where $N(v)$ denotes the set of neighbours of $v$, $e_{uv}$ are edge features and $\phi$ is any function. It is known that there exist non-isomorphic graphs, for which LU-GNNs produce identical embeddings to every node at each layer (Garg et al., 2020). We now note that this result also holds for graphs embedded in 3D space. Let us introduce *Locally Unordered 3D* GNNs (LU3D-GNN) which aggregate embeddings for each node based on the embeddings of the neighbours, as well as their distance to these neighbours. Formally, in LU3D-GNNs, the messages are defined as:

$$m_{u \to v}^{(l-1)} = \phi\left(h_u^{(l-1)}, e_{uv}, \|x_u - x_v\|\right) \tag{7}$$

where $x_v$ is the 3D position of $v$. We have the following result.

**Lemma 1.** *There exist connected non-isomorphic geometric graphs that differ in the number of conjoined cycles, girth, size of the largest cycle and cut-edges that LU3D-GNNs cannot distinguish.*

On the lefthand side of Figure 4, we show two graphs that are not isomorphic and differing in all mentioned properties, but which cannot be distinguished by LU3D-GNNs. For clarity of presentation, we presented the graphs in 2D, but any 3D configuration having all edges of equal length can be used. We prove that they cannot be distinguished by LU3D-GNNs in Appendix E.

In the context of SBDD, we are interested in modelling pairs of graphs corresponding to protein-ligand complexes. We now show that they also pose representational challenges for LU(3D)-GNNs.

Let $G_1 = (V_1, E_1), G_2 = (V_2, E_2)$ be any two graphs and $\mathcal{C}(G_1, G_2)$ denote a *complex* graph, i.e. $\mathcal{C}(G_1, G_2) = (V, E)$, where

$$V = V_1 \cup V_2 \text{ and } E = E_1 \cup E_2 \cup V_1 \times V_2 \tag{8}$$

and, importantly, features for all added edged $e_{uv} \in V_1 \times V_2$ only depend on the features of nodes at their endpoints. We have the following results.

**Proposition 1.**

(i) *If $G_1$ and $G_2$ are indistinguishable for LU-GNNs, then for any graph P, the complex graphs $\mathcal{C}(P, G_1), \mathcal{C}(P, G_2)$ are also indistinguishable for LU-GNNs.*

(ii) *There exist ligand graphs $G_1, G_2$ differing in graph properties listed in Lemma 1, such that for any protein P, the complexes $\mathcal{C}(P, G_1), \mathcal{C}(P, G_2)$ are indistinguishable for models using LU3D-GNNs or LU-GNNs for intra-ligand and intra-protein message passing and LU-GNNs for inter ligand-protein message passing.*

The righthand side of Figure 4 visualizes the proposed statement. The proof can be found in Appendix E. Proposition 1 shows that a scoring model defined in Equation (2) parametrized as a LU-GNN would not differentiate between certain ligand graphs, no matter what the protein context would be. This stems from the fact that we model unlabelled graphs without 3D coordinates and therefore cannot use LU3D-GNNs for inter ligand-protein message passing. We provide a real-world example of a pair of molecules that have different binding affinities to a specific protein pocket, but which are identical from LU-GNN perspective.

## 5.2 GENERALIZATION OF THE SCORING MODEL

We now delve into the theoretical analysis of the generalization capabilities of our proposed scoring model within FastSBDD. Understanding generalization is paramount, as it provides insight into how well our model can perform on unseen data, which is a fundamental aspect of its practical utility in structure-based drug design (SBDD). Our analysis is focused on deriving generalization bounds, which give a theoretical measure of the model's performance across different molecular structures and protein targets. We begin with defining the generalization error.

**Definition 1.** (Generalization error). *Let $f : \mathcal{X} \to \mathcal{Y}$ and $L : \mathcal{Y} \times \mathcal{Y} \to \mathbb{R}_+$ be a loss function. Let $S = \{(x_1, y_1), \ldots, (x_m, y_m)\} \subseteq \mathcal{X} \times \mathcal{Y}$ be a training sample. The generalization error of $f$ is defined as the difference between expected loss and the sample loss:*

$$\mathcal{R}_S(f) = \mathbb{E}L(f(x), y) - \frac{1}{m} \sum_{i=1}^{m} L(f(x_i), y_i), \tag{9}$$

*where the expectation is taken over $(x, y)$ sampled from the underlying data distribution.*

$\varepsilon$-**soft normalized EGNN** As described in Section 3.2, our scoring model first computes an embedding of the protein graph, then concatenates features of ligand's unlabelled graph and passes through an MLP. However, we use a slightly modified version of the EGNN model. As originally introduced, a layer of the EGNN network can be summarized as follows:

$$\mathbf{m}_{u \to v}^{(l-1)} = \phi_e \left( \mathbf{h}_u^{(l-1)}, \mathbf{h}_v^{(l-1)}, \|\mathbf{x}_u^{(l-1)} - \mathbf{x}_v^{(l-1)}\|, e_{uv} \right); \mathbf{m}_v^{(l-1)} = \sum_{u \in \mathcal{N}(v)} \mathbf{m}_{u \to v}^{(l-1)} \tag{10}$$

$$\mathbf{x}_v^{(l)} = \mathbf{x}_v^{(l-1)} + \frac{1}{|\mathcal{N}(v)|} \sum_{u \in \mathcal{N}(v)} \left( \mathbf{x}_v^{(l-1)} - \mathbf{x}_u^{(l-1)} \right) \phi_x(\mathbf{m}_{u \to v}^{(l-1)}) \tag{11}$$

$$\mathbf{h}_v^{(l)} = \phi_h(\mathbf{h}_v^{(l-1)}, \mathbf{m}_v^{(l-1)}), \tag{12}$$

where $\mathbf{h}_v^{(0)}$ and $\mathbf{x}_v^{(0)}$ are initialized with node's features and coordinates respectively and $\phi_x, \phi_e, \phi_h$ are learnable functions, typically MLPs. In our experiments, we replace (11) with

$$\mathbf{x}_v^{(l)} = \mathbf{x}_v^{(l-1)} + \frac{1}{|\mathcal{N}(v)|} \sum_{u \in \mathcal{N}(v)} \frac{\mathbf{x}_v^{(l-1)} - \mathbf{x}_u^{(l-1)}}{\|\mathbf{x}_v^{(l-1)} - \mathbf{x}_u^{(l-1)}\| + \varepsilon} \phi_x(\mathbf{m}_{u \to v}^{(l-1)}). \tag{13}$$

The authors include this version of the model in the official implementation[1], but do not use it in the experiments. With this formulation, we derive the following generalization bound:

**Proposition 2** (Informal). *Let $f$ be the scoring model using $\varepsilon$-soft normalization in the EGNN network. Then for the sample of size $m$, the following holds*

$$\mathcal{R}_S(f) \leq \mathcal{O}\left(\frac{L^2 + k}{\sqrt{m}} \sqrt{\log\left(\frac{m(L+k)}{\varepsilon}\right)}\right), \tag{14}$$

*where $L$ is the number of layers of the EGNN and $k$ is the number of layers in the MLP.*

We note that we make no assumptions about $\varepsilon$ other than $\varepsilon > 0$. If generalization is the primary concern, one can set $\varepsilon = 1$.

*Proof sketch.* The key step is establishing that the model is Lipschitz continuous w.r.t. parameters and determining the Lipschitz constant. We show it by first proving that Lipschitz continuity is preserved under $\varepsilon$-soft normalization. We then use an inductive argument to derive recursive relationship for the constants and determine their growth rate as a function of the depth of the network. Once the Lipschitz constants are established, we leverage results from Learning theory relating the covering number of a function class to the empirical Rademacher complexity, which can subsequently be used to bound the genneneralization error. Please see Appendix F for a detailed formulation and proof. □

**Impact of $\varepsilon$-soft normalization**  Interestingly, when analyzing the generalization bounds of normalized EGNN model vis-à-vis the original formulation, we observe an important difference. Our derivations indicate that, in the absence of normalization, the original EGNN model would exhibit significantly less favorable generalization bounds. Specifically, using the same proof technique, the bound for the EGNN without normalization would be exponential in $L$ as opposed to polynomial. Please see Appendix F.5 for details. This discovery underscores the importance of the normalization component, emphasizing its benefits in both numerical stability and theoretical generalizability.

## 6  CONCLUSION, BROADER IMPACT AND LIMITATIONS

Our research brings to the forefront efficient methodologies for Structure-Based Drug Design (SBDD), advocating for a shift in focus from mere model expressivity to robust generalization. The successes of FastSBDD serve as a testament to the significance of integrating metrics of interest into the optimization process. The unprecedented efficiency and success of FastSBDD not only highlights the potential to reshape the approach to SBDD but also paves the way for streamlining docking software, especially when assessed with widely adopted tools like Vina and Gnina.

However, there are limitations to our findings. While we rely on Vina, the gold standard for evaluation in the SBDD realm, the ideal validation would entail wet lab experiments or more sophisticated molecular dynamics simulations. However, such methods are often prohibitively expensive. Additionally, the very nature of FastSBDD as a generative model, especially with its property optimization capabilities, brings with it risks. As SBDD models become increasingly adept at designing molecules with specific properties, we must remain vigilant of the potential unintended outcomes, since streamlining the design process could accelerate the design of harmful biochemicals.

**Reproducibility statement**  All our experiments are reproducible. We will share our code as well as the trained models under the MIT License on GitHub upon the acceptance of the paper.

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
