# OpenReview forum: "Streamlining Generative Models for Structure-Based Drug Design"
_ICLR.cc/2024/Conference — Submitted to ICLR 2024_

### Official Review · Reviewer_A4Na · 2023-10-29

**Soundness:** 2 fair
**Presentation:** 3 good
**Contribution:** 2 fair
**Rating:** 3
**Confidence:** 3

**Summary:**

The author introduced a novel two-step structure-based drug design method. Unlike the previous models that achieved end-to-end sampling, this approach first created the unlabeled atom graph (no atom type, bond type, and atom coordinates) and then sampled the label. They benchmarked their method with existing SOTA methods like DiffSBDD, Pocket2Mol, and TargetDiff. The results show this method could achieve better performance in terms of molecular property scores and speed.

**Strengths:**

1. The writing is generally great. Contributions and novelty are clear.
2. The motivation is clear and the problem that the paper wanted to address is important.
3. The paper introduced the theoretical analysis.

**Weaknesses:**

Firstly, it is important to note that the method did not actually achieve de-novo drug design, as it sampled the unlabeled ligand graph from an existing dataset. This raises questions about the novelty of the generated drug.

Furthermore, there is a lack of clarity regarding whether bond types are preserved in the unlabeled graph. Even if bond types are not preserved, it is not intuitively clear whether there is enough flexibility for atom type when the graph structure is fixed.

In light of these concerns, the focus on overall diversity is not as meaningful as it could be. It would be more informative to assess and showcase the diversity of generated ligands for each template unlabeled graph. This analysis can reveal the extent of chemical variation achieved by the method, which is a more relevant measure of its potential impact. If the changes between the generated ligands and the template ligands are minor or do not significantly affect the chemical properties, the overall impact of the method may be considerably diminished.

Without more evidence and explanation regarding the mentioned issues, I think comparing the drug repurpose-like generating method with the de-novo generating method is not fair enough. It might be more appropriate to compare it with other drug repurposing methods.

**Questions:**

1. Is the bond type preserved in the template unlabeled graph?
2. Is the diversity of the generated ligands for a given template unlabeled graph good enough? Are they significantly different from the original ligand?

---

> ### Author Response · Authors · 2023-11-14
>
> Thank you for your feedback and constructive questions. We address them below.
>
> *Firstly, it is important to note that the method did not actually achieve de-novo drug design, as it sampled the unlabeled ligand graph from an existing dataset. This raises questions about the novelty of the generated drug.*
>
> This is a good observation. We would like to point out that we compare with other baselines also in terms of chemical diversity (main results Table 1) and obtain comparable results (68% vs 72%), where diversity is defined as 1 -tanimoto similarity as in all other baselines [1, 2].
>
> **Novelty of the generated molecules.** The novelty of generated molecules with our method depends on which variant is used. The FastSBDD-$\mathcal{DR}$ described in Section 3.3.1 does not generate novel molecules, it is included to showcase methods capability for drug repuprosing. The other variants, however, FastSBDD-$\mathcal{ND}$ and FastSBDD-$\mathcal{PO}$ achieve 100% novelty, meaning that none of the generated molecules are present in the training sets. This shows that these two variants do in fact perform de-novo drug design.
>
> *Furthermore, there is a lack of clarity regarding whether bond types are preserved in the unlabeled graph. Even if bond types are not preserved, it is not intuitively clear whether there is enough flexibility for atom type when the graph structure is fixed.*
>
> **Bond types are indeed preserved in the unlabeled graph.** We will make it more explicit in the paper. There is substantial flexibility for atom types with the fixed graph structure. We discuss in more detail below.
>
> *In light of these concerns, the focus on overall diversity is not as meaningful as it could be. It would be more informative to assess and showcase the diversity of generated ligands for each template unlabeled graph. This analysis can reveal the extent of chemical variation achieved by the method, which is a more relevant measure of its potential impact. If the changes between the generated ligands and the template ligands are minor or do not significantly affect the chemical properties, the overall impact of the method may be considerably diminished.*
>
> **Chemical diversity of our generated molecules is significant.** Thank you for pointing this out. There is in fact substantial variability in the chemical space even with fixed 2D unlabelled graph structures.
>
> **New experiment based on your feedback.** In this experiment, we show that by only changing atom types, we can generate hundreds (and in 98% of cases at least a thousand) novel molecules, which are almost as diverse as the ZINC250k dataset itself. Please see the General Response for details.
>
> *Without more evidence and explanation regarding the mentioned issues, I think comparing the drug repurpose-like generating method with the de-novo generating method is not fair enough. It might be more appropriate to compare it with other drug repurposing methods*
>
> **Our scoring model enables both repurposing and new drug generation.** We note that the FastSBDD-$\mathcal{DR}$ variant of our method, designed for drug repurposing, was meant to demonstrate that our method can have wider applications. Due to our scoring model, we are able to perform both drug repurposing as well as novel drug generation with the latter being the primary concern. This is why we benchmark our method against the generative models for SBDD.
>
> Thanks again for your review and excellent questions. In response to your concerns, we have provided additional empirical evidence which we hope has been convincing. We would appreciate it if you could reflect this in an updated score.
>
> [1] Peng, X. et al., Pocket2Mol: Efficient Molecular Sampling Based on 3D Protein Pockets, ICML (2022)
>
> [2] Schneuing A. et al., Structure-based Drug Design with Equivariant Diffusion Models, arXiv preprint arXiv:2210.13695 (2022)

---

> ### Author Response · Authors · 2023-11-20
> **Gentle reminder**
>
> We once again thank the Reviewer for the review and insightful feedback. We believe that our response and especially the new analysis demonstrating chemical variability of molecules with fixed 2D unlabelled structures have addressed the raised concerns and would therefore kindly request re-examination of our work. If any further inquiries or issues persist, we encourage you to bring them to our attention for swift resolution.

---

> > ### Comment · Reviewer_A4Na · 2023-11-21
> >
> > Thanks for providing additional details! They are quite valuable.
> >
> > The new results do address some of my concerns. However, I'd like to keep the score for the following reasons.
> >
> > The performance of the atom-type regeneration process and property optimization process is not good.
> >  a. When comparing FastSBDD-DR and FastSBDD-ND in Table 1, no metric exhibits improvement following the regeneration of atom types.
> >  b. When comparing FastSBDD-DR and FastSBDD-PO in Table 1, there is no significant improvement in any metric except for QED.
> >
> > It appears to me that the key reason for FastSBDD's superiority over baselines is that they use a large unlabeled graph dataset which makes the algorithm actually a drug-repurpose-like method.

---

> > > ### Author Response · Authors · 2023-11-21
> > >
> > > Thank you for this comment and the opportunity to further clarify.
> > >
> > > *When comparing FastSBDD-DR and FastSBDD-ND in Table 1, no metric exhibits improvement following the regeneration of atom types*
> > >
> > > The key difference between these two variants is novelty. 100% of generated drugs by FastSBDD-ND are novel, i.e. not seen in the training set. No improvement in other metrics can be explained by the fact that FastSBDD-ND is not optimized for any particular metric, but rather uses a generative model that matches the training data distribution. Therefore, it is expected that it exhibits similar performance in various properties.
> > >
> > > *When comparing FastSBDD-DR and FastSBDD-PO in Table 1, there is no significant improvement in any metric except for QED.*
> > >
> > > FastSBDD-PO also generates novel molecules as opposed to FastSBDD-DR. The reason that it is the QED metric that improved is that this was what we chose to optimize. We mention in section 3.3.3 that, as a demonstration, we define the property that we wish to optimize as: 5 * QED + SA. If, for example, SA or any other metric is somebody’s primary concern then one can easily change a single line in our code and obtain molecules optimized for that.
> > >
> > > *It appears to me that the key reason for FastSBDD's superiority over baselines is that they use a large unlabeled graph dataset which makes the algorithm actually a drug-repurpose-like method.*
> > >
> > > This is true that we are the first to utilise the database of pre-defined unlabelled structures in the SBDD domain and this indeed results in superiority over baselines. We view this as our contribution rather than a limitation and believe that this is something that SBDD community can find interesting. We see in the main results Table 1 that this does not significantly reduce diversity of generated molecules. We further argued in this rebuttal that fixing unlabelled graphs still allows for great flexibility in the molecular space.
> > >
> > > FastSBDD-ND and FastSBDD-PO models that we propose are not drug repurposing methods. These are generative models that generate unseen molecules in 100% of the cases. We included the FastSBDD-DR variant in the presentation to demonstrate another use case for our scoring model, whenever one wishes to analyze existing drugs in the context of protein-ligand binding. However, we re-emphasize that it is the novel drug generation that is our primary concern and therefore we present two model variants (FastSBDD-ND and FastSBDD-PO) that serve this purpose and compare with methods that also generate new drugs.
> > >
> > > We are also open to moving the FastSBDD-DR variant of the method to a different section to make the distinction more explicit and exclude it from the direct comparison with SBDD baselines if this makes the presentation clearer.
> > >
> > > ---
> > > Hope this addresses your remaining concerns. Please let us know if there's still anything else we can do to persuade you to increase your score.

---

### Official Review · Reviewer_aZds · 2023-10-29

**Soundness:** 1 poor
**Presentation:** 2 fair
**Contribution:** 2 fair
**Rating:** 3
**Confidence:** 4

**Summary:**

This work proposes a two-stage method, FastSBDD, for structure-based drug design. In the first stage, a binding-affinity predictor model is used to infer an unlabeled molecular graph. In the second stage, labels are optimized on the graph to improve the molecular properties, such as QED and SA. The results are considerably good and the inference speed is extremely fast. Additionally, theoretical analyses on representation learning under SBDD setting are also provided.

**Strengths:**

1. This paper introduces metric-aware optimization for generative SBDD settings, which is interesting.
2. Previous methods directly 3D ligands based on give 3D pockets. However, this paper first generates unlabeled graph (topology) based on the 3D pockets, and then generated atom types and positions based on the unlabeled graph independently of proteins. The experimental results are promising.
3. Theoretical analyses are provided which are lacked in previous SBDD works.

**Weaknesses:**

1. The rationality of the two-stage method is suspectable. In general, especially from the perspective of structural biology, the protein-ligand interaction is strongly dependent on atom types, positions. However, in this paper, the only dependence on proteins is used to infer the unlabeled molecular graph where only topology information is used. And the second stage does not involve modelling the protein pockets. This contradicts with the fundamental knowledge of structural biology.
2. The main concern about the experiments is about the fairness of comparison with baselines. First, FastSBDD leverages additional molecular dataset, such as ZINC and ChemBL, while other baselines does not use additional datasets. Second, optimization is condered in FastSBDD. But the baselines are all generative methods. No optimization methods, such as MARS [1] and RGA [2], are included. From my perspective, it is more proper to compare FastSBDD with molecular optimization methods. RGA is a molecular optimization method proposed for SBDD.
3. Besides, many important SBDD baselines are missing, such as DecmopDiff [3].

References:

[1] Xie, Yutong, et al. "Mars: Markov molecular sampling for multi-objective drug discovery." arXiv preprint arXiv:2103.10432 (2021).

[2] Fu, Tianfan, et al. "Reinforced genetic algorithm for structure-based drug design." Advances in Neural Information Processing Systems 35 (2022): 12325-12338.

[3] Guan, Jiaqi, et al. "DecompDiff: Diffusion Models with Decomposed Priors for Structure-Based Drug Design." (2023).

**Questions:**

The main questions is about how much protein-ligand interation information is included in unlabeled graphs. More analyses are needed.

---

> ### Author Response · Authors · 2023-11-14
>
> Thank you for your feedback and comments. We address them below.
>
> *The rationality of the two-stage method is suspectable. In general, especially from the perspective of structural biology, the protein-ligand interaction is strongly dependent on atom types, positions. However, in this paper, the only dependence on proteins is used to infer the unlabeled molecular graph where only topology information is used. And the second stage does not involve modelling the protein pockets. This contradicts with the fundamental knowledge of structural biology.*
>
> This is a great point. We agree that this model definition and results are counter-intuitive and we believe that therefore it is especially important to publish such findings. Please see our General Response where we re-emphasize points made in the paper and discuss this model definition.
>
> *The main concern about the experiments is about the fairness of comparison with baselines. First, FastSBDD leverages additional molecular dataset, such as ZINC and ChemBL, while other baselines does not use additional datasets*
>
> This is a good point. However, it is important to note two things. First, the additional datasets, such as ZINC and ChemBL only contain molecules, they do not contain molecule-protein pairs indicating strong binding of a given molecule to a given protein pocket as in the CrossDocked dataset used by SBDD
> methods.
>
> Second, our method is conceptually different than other benchmarks. Our approach with utilizing additional molecular datasets is novel in the SBDD domain and a part of our contribution. It is unclear how other baselines could benefit from these additional data.
>
> *Second, optimization is condered in FastSBDD. But the baselines are all generative methods. No optimization methods, such as MARS [1] and RGA [2], are included. From my perspective, it is more proper to compare FastSBDD with molecular optimization methods. RGA is a molecular optimization method proposed for SBDD.*
>
> Thank you for pointing this out. Please see our general response, where we discuss comparison with optimization-based methods.
>
> *Besides, many important SBDD baselines are missing, such as DecmopDiff [3]*
>
> In the comparison, we have included a recent SBDD baseline: TargetDiff (ICLR 2023). DecompDiff was published in a peer reviewed venue in June 2023 (ICML 2023), making it too recent (after 28.05.2024) to be expected to include in the analysis as per ICLR 2024 guidelines. DecompDiff uses a different software for estimating binding affinity (as indicated e.g. by different results for Pocket2Mol than originally reported), so it is not straightforward to compare with our method without recomputing the numbers ourselves. We will try to do that before the end of the discussion phase.
>
> **Strong performance compared to DecompDiff.** DecompDiff takes on average 6189s to generate 100 molecules per protein pocket, making it 1.8x slower than the slowest baseline TargetDiff. Furthermore, as we argue in the paper, we believe that the ultimate goal of SBDD should be to focus on both the binding affinity and other molecular properties. DecompDiff demonstrates subpar performance in terms of most commonly used molecular properties:
> * QED: FastSBDD ∈ [0.61, 0.8]; DecompDiff = 0.45
> * SA: FastSBDD ∈ [0.69, 0.78]; Decompdiff = 0.61
> Other metrics like LogP or Lipinski are not reported.
>
> *The main questions is about how much protein-ligand interation information is included in unlabeled graphs. More analyses are needed.*
>
> Empirically, we have shown that using the unlabelled molecular graphs alone is enough to predict binding scores with accuracy high enough to outperform many SBDD baselines. This is an interesting and unexpected result, especially given that this behaviour was exhibited not only by the Vina Score, but also Gnina, a newer and reportedly, significantly more accurate docking software (that we mention in Appendix A.4).
>
> We believe that this opens up further questions about the design of SBDD methods and development of docking software, but an extensive analysis from the structural biology point of view is not in the scope of this study.
>
> Thanks again for your feedback. We hope that our response and clarifications helped clear your concerns and you would reconsider and amend your original score.

---

> ### Author Response · Authors · 2023-11-20
> **Gentle reminder**
>
> Thank you once again for your detailed review and constructive feedback. We feel that our response and especially the clarifications of the baselines comparisons have sufficiently addressed your concerns and would thus kindly request a re-evaluation of our submission. Should there be any other concerns, questions or requests for clarifications, we are eager to respond.

---

### Official Review · Reviewer_zMwy · 2023-11-01

**Soundness:** 2 fair
**Presentation:** 3 good
**Contribution:** 2 fair
**Rating:** 5
**Confidence:** 4

**Summary:**

This paper proposes a new generative model named FastSBDD for structure-based drug design. It disentangles the ligand molecule generation into the unlabelled molecular graph generation and the atom type generation. The unlabelled molecular graph generation module is actually a sampler, which samples unlabelled graphs from ligand database.  The authors build a binding affinity prediction model to select unlabelled graphs with best predicted affinities. The atom type generation module can be a sampler, MoFlow based generator or a generator combined with property filters. The authors perform experiments on Crossdocked2020 and show that their model can achieve better performance with much less time cost. Moreover,  the authors provide theoretical analysis to justify that FastSBDD can generalize well.

**Strengths:**

- The proposed method is simple and effective.
- The writing is clear and easy to follow.
- The code is provided.

**Weaknesses:**

- The proposed disentanglement doesn't make much sense to me.  I wonder whether the authors redock molecules with AutoDock Vina in their preliminary experiments (Sec 3.1 and Appendix A). I suspect that changing initial 3D conformation can also achieve a high $R^2$ of Vina score is caused by redocking. Otherwise, the Vina score should have a large change.
- The assumption justification is needed about why the unlabelled graph sampler could be independent of protein
- The proposed method has many filtering operations (filter high-affinity molecules with binding affinity scoring model, filter high QED+SA molecules in the atom type assignment phase) during the generation process. It would be fairer to compare with optimization-based models or compare with generative models and apply a similar property screening operation as post-processing.
- The proposed method can not generate novel 2D graph structures. I'm not sure whether it's a real limitation. If the authors can justify all 2D graph structures can be covered by the ligand database, it's also fine.

**Questions:**

- How accurate is the binding affinity scoring model? Typically, the model utilized 3D information would be more accurate, but the proposed scoring model only uses the unlabelled 2D molecular graph.
- In theoretical analysis, LU-3DGNN can distinguish G1, G2 with protein-ligand inter message passing. Does it mean considering ligand 3D information will lead to a better model?

---

> ### Author Response · Authors · 2023-11-14
>
> We thank the Reviewer for a careful review and raising important questions. We address them below.
>
> *The proposed disentanglement doesn’t make much sense to me. I wonder whether the authors redock molecules with AutoDock Vina in their preliminary experiments (Sec 3.1 and Appendix A). I suspect that changing initial 3D conformation can also achieve a high $R^2$ of Vina score is caused by redocking. Otherwise, the Vina score should have a large change.*
>
> **Role of redocking.** Yes, redocking is precisely the reason why coordinates have marginal effect on the Vina Score. The Vina Score that we and other baselines use [1, 2] for estimating docking scores does use redocking. We make it explicit in Eq (16) in Appendix A.1 where we provide a detailed definition of
> the Vina Score.
>
> *The assumption justification is needed about why the unlabelled graph sampler
> could be independent of protein*
>
> **Justification for assumption about conditional independence.** We assume the Reviewer meant ”why the **atom sampler** could be independent of protein”, because the unlabelled graph sampler is not independent of the protein. Eq (3) explicitly shows the dependence of the unlabelled graph sampler
> on the protein.
>
> The conclusion that the **atom sampler** can be effectively taken to be independent of the protein is drawn from the experiments mentioned in Section 3.1 and detailed in Appendix A. This choice is motivated by empirical findings based on the software approximating the docking scores. Please refer to our General Response for more details.
>
> *The proposed method has many filtering operations (filter high-affinity molecules with binding affinity scoring model, filter high QED+SA molecules in the atom type assignment phase) during the generation process. It would be fairer to compare with optimization-based models or compare with generative models and apply a similar property screening operation as post-processing.*
>
> This is a great point. We view our method as bridging the gap between the generative optimization-based models. To the best of our knowledge, we are the first to introduce the explicit optimization of binding (and other molecular properties) to a generative model without having to rely on expensive computations. On the contrary, our method is lightning fast compared to baseline generative models for SBDD, which are themselves faster than optimization based models. Please see our General Response for more details.
>
> **Post-hoc screening would result in an even more significant relative speedup.** Regarding screening predictions of generative models as post-processing, the scoring model that we use for filtering is proprietary and is part of our contribution. If we compared other baseline models and filtered their predictions using the Vina software this would come at a cost. Scoring a single protein-ligand pair takes on average 16 seconds. Assuming that we generated double the number of predictions, scored them all with Vina and filtered out bottom half, this would quadruple the runtime of the fastest baseline, which already is two orders of magnitude slower than the slowest variant of our method. It is also not obvious whether optimization for binding alone would not hurt other molecular properties. We demonstrate with our method that we can do both: outperform baselines in terms of both binding and other molecular properties.
>
> *The proposed method can not generate novel 2D graph structures. I’m not sure whether it’s a real limitation. If the authors can justify all 2D graph structures can be covered by the ligand database, it’s also fine.*
>
> Thank you for the opportunity to clarify this. As we mention in Section 3.2, it is possible to extend the method to allow for sampling entirely new 2D graph structures, but we decide not to do that. We note that there is substantial variability in the molecular space even with a fixed set of 2D unlabelled graph structures. This is supported by the fact that our method has comparable chemical diversity to other methods as shown in Table 1.
>
> **Diversity of the generated molecules.** To demonstrate the diversity of the molecular space with fixed 2D unlabelled graph structures, we conducted an additional experiment. We showed that we can generate hundreds (and in 98% of the cases at least a thousand) of novel and chemically diverse molecule for a fixed 2D unlabelled structure. Please see our General Response for more details.
>
> Regarding the novelty of generated molecules, it is true that we cannot generate unseel 2D graph structures, but the FastSBDD-$\mathcal{ND}$ and FastSBDD-$\mathcal{PO}$ variants of our model achieve 100% novelty, meaning that none of the generated molecules are present in the train set.

---

> > ### Author Response · Authors · 2023-11-14
> >
> > *How accurate is the binding affinity scoring model? Typically, the model utilized 3D information would be more accurate, but the proposed scoring model only uses the unlabelled 2D molecular graph*
> >
> > **Accuracy of the binding affinity scoring model.** The binding affinity scoring model achieves 71% correlation with the ground truth on the validation set.
> >
> > As we mention in section 3.1 and provide more details in Appendix A, there is marginal effect of the exact 3D configuration of the molecule on the Vina Score as it performs redocking as part of its estimation of the docking scores. Of course, the scoring model cannot be 100% accurate, because it does not use atom type information. This is done on purpose, because (1) it allows for sampling novel drugs and (2) we show empirically that this is enough to outperform
> > existing methods in terms of predicted binding affinity.
> >
> > *In theoretical analysis, LU-3DGNN can distinguish G1, G2 with protein-ligand inter message passing. Does it mean considering ligand 3D information will lead to a better model?*
> >
> > **Effect of incorporating 3D information on expressivity and generalization.** This is a great observation. LU-3DGNNs are indeed more expressive than LU-GNNs, because they use more information and in general can model 3D graphs better. That said, in this project, we have shown empirically that 3D information does not contain useful information for predicting the binding scores (because of redocking) and therefore our design choice to model unlabelled graphs without 3D information was motivated by empirical findings. In such scenarios, the increase in representational capacity (i.e., expressivity) using powerful models that accommodate the aditional spatial information can lead to worse generalization so the predictive performance can suffer, i.e., we may not get a better model. In fact, this may explain why approaches such as the one proposed in this work that try to balance expressivity and generalization tend to perform better than the existing approaches that focus solely on expressivity at the expense of generalization.
> >
> > Thanks again for your feedback and questions. We trust that our detailed response, along with the new empirical data provided, has adequately resolved your queries, and we will thus greatly appreciate your stronger support for this work with a revised rating.
> >
> > [1] Peng, X. et al., Pocket2Mol: Efficient Molecular Sampling Based on 3D Protein Pockets, ICML (2022)
> >
> > [2] Schneuing A. et al., Structure-based Drug Design with Equivariant Diffusion Models, arXiv preprint arXiv:2210.13695 (2022)

---

> ### Author Response · Authors · 2023-11-20
> **Gentle reminder**
>
> We would like to thank the reviewer again for the thorough review and insightful questions. We believe that our response together with the additional analysis conducted have addressed the concerns and would kindly request re-evaluation of our work. If there are any further concerns or questions, we are happy to address them.

---

> > ### Comment · Reviewer_zMwy · 2023-11-21
> >
> > Thank you for the authors' response, however, my concerns still remain. The authors use Vina with redocking to perform preliminary experiments and end up the core motivation of this work: "a considerable portion of the variability in binding affinity, as quantified by the Vina score, can be attributed to the information contained within the unlabeled molecular graph itself". I think the logic is problematic. The redocking procedure totally ignores the initial 3D information, which doesn't mean the 3D information is not important in determining the binding affinity. In addition, I still don't think the comparison with de-novo SBDD baselines is fair, since they are not designed to optimize some molecular properties and they can perform de-novo design but the proposed method in this work can not (due to fixed unlabelled 2D graph). I think the authors should compare with other molecular optimization models in SBDD tasks (such as [1][2] etc. ) to make this work more solid.
> >
> > [1] Fu, T., Gao, W., Coley, C., & Sun, J. (2022). Reinforced genetic algorithm for structure-based drug design. Advances in Neural Information Processing Systems, 35, 12325-12338.
> >
> > [2] Yang, Y., Ouyang, S., Dang, M., Zheng, M., Li, L., & Zhou, H. (2021). Knowledge Guided Geometric Editing for Unsupervised Drug Design.

---

> > > ### Author Response · Authors · 2023-11-22
> > > **Response to Reviewer zMwy 1/2**
> > >
> > > Thank you for this comment. We address your points below.
> > >
> > > *The authors use Vina with redocking to perform preliminary experiments*
> > >
> > > Yes, we perform the preliminary experiments with Vina with redocking, which is the gold standard for evaluation in the SBDD space [3, 4, 5]. The two suggested methods by the Reviewer [1, 2] also evaluate their performance using Vina with redocking. We would like to re-emphasize that the findings of our preliminary experiments were reinforced by the newer software Gnina (Appendix A.4).
> > >
> > > *I think the logic is problematic. The redocking procedure totally ignores the initial 3D information, which doesn't mean the 3D information is not important in determining the binding affinity*
> > >
> > > We do not claim that 3D information is not important in determining the binding affinity. As we describe in the Appendix A.1, Vina (with redocking) performs optimization over molecule’s 3D pose. The Vina score obtained with this procedure can be interpreted as “binding potential”, i.e. molecule’s binding affinity in its optimal pose.
> > >
> > > *In addition, I still don't think the comparison with de-novo SBDD baselines is fair, since they are not designed to optimize some molecular properties and they can perform de-novo design but the proposed method in this work can not (due to fixed unlabelled 2D graph)*
> > >
> > > We address this in three parts. First, we take this opportunity to emphasize that **during inference our method only takes the protein pocket as input**. We do not use the 2D graph structure of the ground truth molecule corresponding to the protein pocket. Our model samples the 2D unlabelled graph from a database, where the database is independent of the protein pocket.
> > >
> > > Second, we think it is not straightforward to classify our method as de-novo or not. The approaches to SBDD can roughly be divided into three categories:
> > > * Virtual screening - scanning a database of existing molecules and choosing the best one;
> > > * Optimization of known ligands - starting from a ligand known to bind to a given protein and modifying it to increase binding affinity;
> > > * De-novo design - generating a new molecule only based on a protein pocket.
> > >
> > > Our method (FastSBDD-ND and FastSBDD-PO) does not perform virtual screening, because it generates new and unseen molecules and it does not score existing molecules (it only scores existing 2D unlabelled structures which are not themselves molecules). It also does not perform optimization of known ligands, because it does not require or use the ground truth ligand during generation. It is debatable whether it starts “from scratch” to be classified as purely de-novo, because it does start the generation from an unlabelled structure.
> > >
> > > Finally, we believe that we are solving the same problem as the baseline methods, namely: given an unseen protein pocket, generate candidate molecules that are likely to bind. The Reviewer correctly points out that the prevailing paradigm in the SBDD domain is to learn the distribution of molecules (ligands) conditioned on protein pockets purely from data without any explicit property or binding optimization. What we offer in our paper is a different perspective of the SBDD problem. We view the introduction of explicit optimization of binding, i.e. metric-aware learning, as one of our contributions and empirically show that this allows to outperform baselines and still benefit from significant computational savings. Having said that, we believe that the comparison is fair, because all things considered we are solving the same problem as the baselines, i.e. generating new molecules likely to bind to a given protein pocket, but take a different route than the existing methods. The outcome has the potential for better practical utility as it is generally desirable for molecules to not only have good binding affinity, but also QED, SA etc.
> > >
> > > *I think the authors should compare with other molecular optimization models in SBDD tasks (such as [1][2] etc. ) to make this work more solid.*
> > >
> > > As we mention in the General Response, our method is much faster than optimization models in the SBDD domain. For example, RGA [1] suggested by the reviewer needs 3.2 hours of **GPU** time to generate 100 molecules for a protein pocket making it 100x times slower than our slowest variant, which needs 115 seconds of **CPU** time. Including [1] as a baseline in our comparison would require generating molecules for 100 test protein pockets, meaning 13 days of GPU time. It it difficult to compare with [2], as the runtime is not reported and the implementation is not published.

---

> > > > ### Author Response · Authors · 2023-11-22
> > > > **Response to Reviewer zMwy 2/2**
> > > >
> > > > [1]  Fu, T., Gao, W., Coley, C., & Sun, J. (2022). Reinforced genetic algorithm for structure-based drug design. Advances in Neural Information Processing Systems, 35, 12325-12338.
> > > >
> > > > [2] Yang, Y., Ouyang, S., Dang, M., Zheng, M., Li, L., & Zhou, H. (2021). Knowledge Guided Geometric Editing for Unsupervised Drug Design.
> > > >
> > > > [3] Peng, X. et al., Pocket2Mol: Efficient Molecular Sampling Based on 3D Protein Pockets, ICML (2022).
> > > >
> > > > [4] Schneuing, A. et al., Structure-based Drug Design with Equivariant Diffusion Models, arXiv preprint arXiv:2210.13695 (2022).
> > > >
> > > > [5] Luo, S. et al., A 3d generative model for structure-based drug design, NeurIPS (2021)
> > > >
> > > > ---
> > > > We hope this resolves the mentioned concerns If there is more we can do to encourage a higher rating from you, please do not hesitate to inform us.

---

### Official Review · Reviewer_jdjh · 2023-11-06

**Soundness:** 3 good
**Presentation:** 2 fair
**Contribution:** 3 good
**Rating:** 5
**Confidence:** 4

**Summary:**

This paper proposes a two-pronged approach to generate molecules for specific protein target: (1) learn an unlabeled molecular graph (2) optimize conditioned on desired molecular properties. The method shows performance improvement.

**Strengths:**

This paper dicusses the issue with 3D molecule generation for specific protein targets and proposes the method to solve it. The story line is clear.

Empirical results look strong, besides the trained model is smaller in size, which demonstrate the efficiency of proposed method.

**Weaknesses:**

The writing of this paper is not very clear. For example, one of the assumption is the conditional independence in Eq (1), however this equation is not self-inclusive. It's not clear what the model assumptions are and what the conclusions are.

My major concerns is that I think the methodology lacks novelty. Most of section 3 is spent on describing what architectures are used to model certain conditional probabilities. It's not clear what is this paper's techinical contribution.

Section 5 describes theoritical analysis, but I don't see how it is connected to the main claim of this paper. For example, section 5.2 talks about generalization bound, the author may want to show that the model has good generalization ability with respest to some terms (L, m, k) in Eq (14).

**Questions:**

Please refer to *weaknesses* section.

Eq (1), I think the term "disentengled" is inappropriate here. It's just conditional independence, disentenglement usually refers to something else.

---

> ### Author Response · Authors · 2023-11-14
>
> Thank you for your feedback. We address your concerns below.
>
> *The writing of this paper is not very clear. For example, one of the assumption is the conditional independence in Eq (1), however this equation is not self-inclusive. It’s not clear what the model assumptions are and what the conclusions are.*
>
> Eq (1) emphasizes that our model assumes that atom labels $\mathbf{a}^M$ are conditionally independent from the protein graph $G^P$ given the unlabelled graph $U^M$ . As we mentioned in this section, the conclusion is that we can independently train the unlabelled graph sampler model, which is conditioned on the protein context and the atom sampler, which is only conditioned on the unlabelled graph. Empirically, we verify that this formulation allows for a significant speed-up in computation without sacrificing binding performance.
>
> *My major concerns is that I think the methodology lacks novelty. Most of section 3 is spent on describing what architectures are used to model certain conditional probabilities. It’s not clear what is this paper’s techinical contribution*
>
> **Recap of our contributions.** We list our key contributions below:
> * **(Novelty of our metric aware optimization.)** We are the first to introduce drug generation with metric tuning in the Structure-based drug design domain (SBDD). Previously, generative models were trained to match the data distribution, but evaluated against metrics such as binding affinity, which are not taken into account during training.
> * **(Practical impact.)** Our SBDD method is both very accurate while also up to 1000x faster.
> * **(Theoretical contribtutions.)** We include novel theoretical analysis of representational capacity and generalization in the context of SBDD.
>
> *Section 5 describes theoritical analysis, but I don’t see how it is connected to the main claim of this paper. For example, section 5.2 talks about generalization bound, the author may want to show that the model has good generalization ability with respest to some terms (L, m, k) in Eq (14)*
>
> **Relevance of our theoretical analysis with respect to rest of the paper.** Our approach differs from others in the SBDD domain in that we show that with significantly reduced models we can outperform other methods whilst being much more computationally efficient. Apart from the empirical evidence that we provided, we include a theoretical analysis, which emphasizes the point that increasing model’s size
> * does not solve some representational challenges, as there are graphs that cannot be modeled accurately regardless of the depth of the model (Section 5.1)
> * comes at a cost of generalization. The larger the models are the less likely
> they are to generalize well to unseen data distributions. (Section 5.2)
>
> To strengthen our generalization claim, we provide empirical evidence at the bottom of section 4 that our model generalizes well when we change the molecular dataset that we use for sampling.
>
> *Eq (1), I think the term ”disentengled” is inappropriate here. It’s just conditional independence, disentenglement usually refers to something else.*
>
> Thank you for your suggestion. We agree and will change the wording to conditional independence.
>
> Thanks again for all your questions. We hope that our response and clarifications have sufficiently addressed your concerns and you will consider updating your score.

---

> > ### Author Response · Authors · 2023-11-20
> > **Gentle reminder**
> >
> > Thank you again for your insightful review and valuable feedback. We believe that we have clarified the claims in the paper and addressed the mentioned concerns. We kindly request a re-evaluation of our submission. Should there be any further points that require clarification or improvement, please know that we are fully committed to addressing them promptly.

---

### Author Response · Authors · 2023-11-14
**General Response**

We thank the reviewers for their insightful comments and suggestions to improve this work, and to the area, program, and general chairs for their great service to the community.

**Positive aspects.** The reviewers variously acknowledged many strengths of the paper, including, clear story line (jdjh) and strong empirical performance and efficiency, effectiveness, and promise of the proposed approach including small trained model size and extremely fast inference (jdjh, zmWy, aZds); code availability (zMwy); clear contributions and novelty of metric-aware optimization as well as theoretical analyses in generative SBDD settings (aZds, A4Na); clarity of writing (zMwy, A4Na); and motivation and importance of the problem being tackled (A4Na).

**Concerns and Questions.** The reviewers also provided thoughtful feedback, especially regarding the chemical variability of molecules sharing the same 2D graph structure and the separation of the unlabelled graph sampler and the atom sampler. We summarize below their main concerns and questions, and detail the additional analyses conducted by us to address these concerns. We will include all these results in the final version.

### Chemical variability with fixed 2D unlabelled graphs
In our method, when generating novel molecules, we use a set of existing molecular graphs to define possible 2D unlabelled graph structures and only generate atom types. Two Reviewers (A4Na and zMwy) raised a very good point. Namely, *is there sufficient variability in the chemical space if the set of
2D unlabelled graph structures is fixed?*

**Strong empirical evidence from new experiments.** To answer this question, we performed the following experiment. We randomly sampled 100 molecules from the ZINC250k dataset. For each molecule, we used its template unlabeled graph to generate novel valid molecules using the procedure described in Appendix A.3. Whenever we generated a duplicate molecule we tried again. We terminated the procedure when we generated 1000 unique molecules or reached the limit of 10 failed trials due to duplicate generation.

For 98 of these 100 molecules we successfully generated 1000 unique novel molecules sharing the identical 2D unlabelled graph including bond types. For the remaining 2, the procedure terminated after generating 665 and 566 unique novel molecules respectively. This shows that in 98% of the cases we are able to create at least 1000 novel valid molecules while keeping the unlabelled graph
fixed.

Furthermore, we computed the average Tanimoto similarity between the original molecule and the newly generated ones sharing the unlabelled graph. We found that the average Tanimoto similarity was 0.28±0.08 across the 100 sampled molecules from ZINC250k. We compared this number with the average pairwise Tanimoto similarity of the 100 molecules original molecules, which was 0.24 (within one standard deviation). We also contrast this number with the 0.85 similarity threshold, above which molecules have a high probability of having the same activity [1].

This shows that, even for a fixed 2D molecular graph with defined bond types, we can define a large number of unique and valid molecules whose similarity to the original molecule is comparable with the pairwise molecular similarity in the ZINC250k dataset and thus proving the substantial chemical variability.

[1] Patterson DE et al. Neighborhood behavior: a useful concept for validation of ”molecular diversity” descriptors. J Med Chem. (1996)

### Separation of unlabelled graph sampler and atom sampler
Our method for generating molecules with high probability to bind to a given protein relies on a two-stage process: first find unlabelled graph structures optimized for binding and then infer atom types independently of the protein context. Two Reviewers (zMwy and aZds) raised concerns around the proposed approach to separate the two components.

We take this opportunity to re-emphasize points made in the paper.
* The proposed model decomposition is motivated by empirical study conducted with the Vina software, which showed that substantial amount of information needed to predict binding scores is contained in the unlabelled 2D graph structure (Section 3.1 and Appendix A).
* We chose The Vina software as it is the most widely used in the SBDD domain, however a repeated experiment with Gnina, a reportedly significantly more accurate software, yielded very similar results (Appendix A.4), showing that this is not merely an artifact of one specific software, but a more widespread phenomenon.
* *Atom types do impact the docking scores*, but the unlabelled 2D graph structure alone is enough to predict the docking scores with enough accuracy to outperform existing SBDD baselines.
* We make no wet lab experiments of the actual biological process of protein-ligand binding.

---

> ### Author Response · Authors · 2023-11-14
> **General Response continued**
>
> We agree with Reviewer aZds that these findings are very unexpected and counter-intuitive from the perspective of structural biology. We believe that therefore it is especially important to publish such findings as they have the potential to inspire progress both in the SBDD domain as well as the development of more accurate docking software. Wet lab experiments and the analysis from the structural biology point of view were out of scope of this study.
>
> ### Comparison with optimization based methods
> Our method for generating a molecule likely to bind to a protein begins with sampling proposal unlabelled 2D graph structures and filter out those unlikely to bind according to our scoring model. Reviewers zMwy and aZds suggested that it may be more appropriate to then compare with optimization based methods rather than generative models for SBDD. We address this in two parts.
>
> **Novelty.** First, to the best of our knowledge, we are the first to introduce explicit optimization for protein-ligand binding in a generative model for SBDD. Further, we demonstrate how our model definition allows for simultaneous optimization of binding and other molecular properties. The scoring model is an integral part of our method.
>
> **Massive computational speedup.** Second, the optimization based methods are very computationally expensive. The ones proposed by Reviewer aZds, MARS and RGA, require 3-4 hours of GPU computations to generate predictions for a single protein pocket compared to 2 minutes of **CPU** time for all three variants of our method combined. To even include both these benchmarks in the comparison, one needs to generate predictions for 100 protein pockets, requiring 1 month of GPU computations. A significant advantage of our method is its ability to very quickly generate molecules likely to bind to a given protein without even needing a GPU.

---

### Meta-Review · Area_Chair_xaWx · 2023-12-06

**Metareview:**

The paper introduces a methodology for structure-based drug design. Reviewers found significant issues in the paper in its current form that were not addressed during the rebuttal phase. Among them, the most important issues concerned lack of clarity, lack of technical novelty, and the experimental design. These issues were not sufficiently cleared during the rebuttal phase. As such I have to recommend rejection at this stage.

**Justification For Why Not Higher Score:**

Lack of clarity was the biggest issue.

**Justification For Why Not Lower Score:**

N/A

---

### Decision · Program_Chairs · 2024-01-16

Reject